# The sand fly (Diptera: Psychodidae) fauna of the urban area of Lassance, Northeast Minas Gerais, Brazil

**Gabriel Barbosa Tonelli, Camila Binder, Victoria Laporte Carneiro Nogueira, Marina Henriques Prado, Gabriela Gonçalves Theobaldo, Aldenise Martins Campos, Carina Margonari de Souza, José Dilermando Andrade Filho** ⬡ *

Grupo de Estudos em Leishmanioses – Instituto René Rachou – FIOCRUZ Minas – Belo Horizonte, Minas Gerais, Brasil

* jose.andrade@fiocruz.br

## Abstract

The present study aimed to check the sand flies' fauna on the municipality of Lassance, Minas Gerais, Brazil and detect the presence of *Leishmania* DNA on the female captured and determine the risk areas of the municipality. Sand flies were collected monthly from May 2018 to April 2019 using automatic light traps for 3 consecutive nights. Eight houses were selected as sample points due its previous reports of tegumentary leishmaniasis and/or canine leishmaniasis. The sand fly's fauna found on the present study it's represented by several medical importance species and the most abundant species found were *Lutzomyia longipalpis* (77.09%) and *Nyssomyia intermedia* (10.06%). *Leishmania infantum* DNA was detected in a pool of *Lu. longipalpis* resulting on a 2.81% of infection rate. By the frequency of the two most abundant species on this study, we developed a risk area map and it draws attention to sample point 6 due to disparate abundance of sand flies at this site (81.81%). Statistical overview shows *Lu. longipalpis* as dominant species and, still, Non-Metric Multidimensional Scaling analysis reveal high similarity on fauna's diversity on the study area. Our findings suggest that the diversity of sand flies from the municipality of Lassance may promote the circulation of *Leishmania infantum* parasites putting in risk the habitants and other mammal's species. Still, our study reinforces the necessity of specific studies focused on breed sites of phlebotomine and its' ecology to expand the knowledge about the behaviour of this group of insects applying directly to leishmaniases' epidemiology.

## Introduction

Sand flies are widely-distributed insects in the world. They include species that are highly adapted to a great diversity of environments and that are also involved in the transmission of parasites of the genus *Leishmania*, which infect several mammals, including humans, causing leishmaniasis. These insects are more abundant in the Neotropical region, with a greater number of species whose densities fluctuate with climatic seasons [1].

**Data Availability Statement:** All relevant data are within the manuscript.

**Funding:** This study was supported by Fundação de Amparo à Pesquisa do Estado de Minas Gerais

(www.fapemig.br, PPM-00792-18, awarded to JF), Conselho Nacional de Desenvolvimento Científico e Tecnológico (www.cnpq.br, 303680/2020-2, awarded to JF. Additionally, this study was partially supported by the Coordination for the Improvement of Higher Education Personnel (Coordenação de Aperfeiçoamento de Pessoal de Nível Superior - CAPES) (www.capes.gov.br, Finance Code 001, awarded to GT). The funders had no role in study design, data collection and analysis, decision to publish, or preparation of the manuscript.

**Competing interests:** The authors have declared that no competing interests exist.

Brazil is a country with a large territorial extension, with the presence of five biomes, which explains the large number of sandfly species found in its territory [2]). The state of Minas Gerais is endemic for leishmaniasis, comprising three of these biomes (Caatinga, Atlantic Forest and Cerrado). Some studies on sand flies carried out in Minas Gerais, in urban, rural and wild areas, show a significant difference between richness and abundance between these environments, always with the prevalence of *Lutzomyia longipalpis* in the urban environment [3–5].

About the biomes, Cerrado is prevalent in the Brazilian state of Minas Gerais (MG), including its Northern Region, but the biome has been experiencing an increasing rate of deforestation due to disordered population growth, which can lead to a change in the epidemiology of leishmaniasis, with the migration of vectors and reservoirs to the urban environment. Leishmaniasis is known to be a disease of wild origin that puts human health at risk when contact is made with areas of its occurrence [6,7].

The municipality of Lassance, in the Northern Region of Minas Gerais, is in an endemic area of leishmaniasis where 19 cases of America tegumentary leishmaniasis (ATL) and two cases of visceral leishmaniasis (VL) were reported between 2010 and 2019 [8]. Studies carried out in both rural and wild environments of this municipality obtained data on phlebotomine fauna, including the detection of *Leishmania* sp. and the description of a new species [9–14]. Several cases of canine leishmaniasis have been recently observed in the area (unpublished data), yet no monitoring with regard to the vectors of leishmaniasis is being carried out in the local urban area of Lassance.

Thus, the present study aimed to survey the sand fly fauna in the urban area of Lassance, MG, and investigate the presence of *Leishmania* DNA in the collected sand flies. The results will enable us to indicate to health authorities of the municipality the areas of highest risk.

## Methods

### Study area

The study area is located in the Northern Region of the state of Minas Gerais (17 ° 53 ’13 "S, 44 ° 34’ 40" W) (Fig 1), on the banks of Rio das Velhas. Lassance encompasses an area of 3214 km$^2$ and has 6,554 human inhabitants. Local economic activities involve agriculture, fishing and livestock.

The municipality of Lassance is situated in the microregion of Pirapora. It is situated in an area of flat relief, predominantly occupied by the Cerrado biome, and has 81% afforestation on its public streets [15] (Fig 2). Lassance has historical and scientific significance as the place where Chagas disease was discovered in 1907 by Carlos Ribeiro Justiniano Chagas, a young sanitary doctor from the Oswaldo Cruz Institute who was brought there to control malaria in the northern village of Minas [16].

### Sand fly sampling

Eight sampling points were selected throughout the urban area of Lassance (Fig 3) according to information obtained by the local municipal health department about residences involved in canine and/or human cases of leishmaniasis. An HP-type light trap was used at each sampling point with a sampling effort of 72 hours per month from May 2018 to April 2019. Climatic data (average temperature and rainfall) for the sampling period were acquired from the National Institute of Meteorology (NIMET). The sampling points were characterized according to the presence or absence of factors that may contribute to the presence of sand flies, as shown in Table 1.

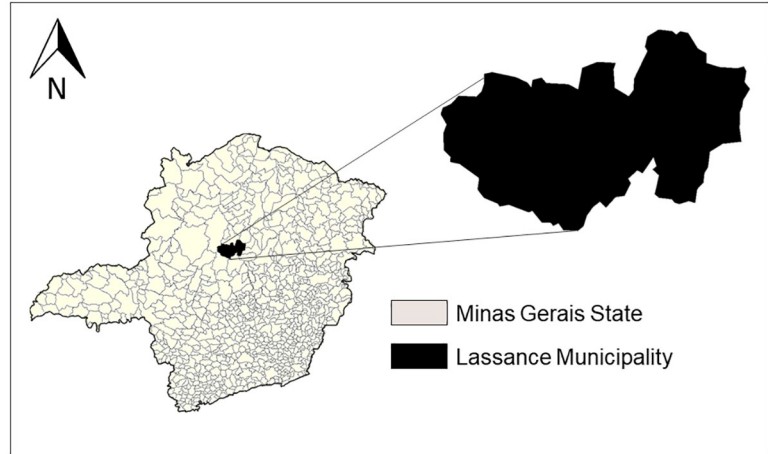

**Fig 1. Location of the municipality of Lassance in the state of Minas Gerais, Brazil.**

Collected sand flies were stored in tubes containing 70% alcohol. In the laboratory, the flies were separated by sex, prepared in Berlese liquid and mounted on microscope slides with a coverslip. The entire body of male individuals was mounted while only the head and the last two abdominal segments were mounted for females with the remainder of the body being separated for molecular analysis. The insects were identified following the classification of Galati [17] and the genera abbreviations followed Marcondes [18].

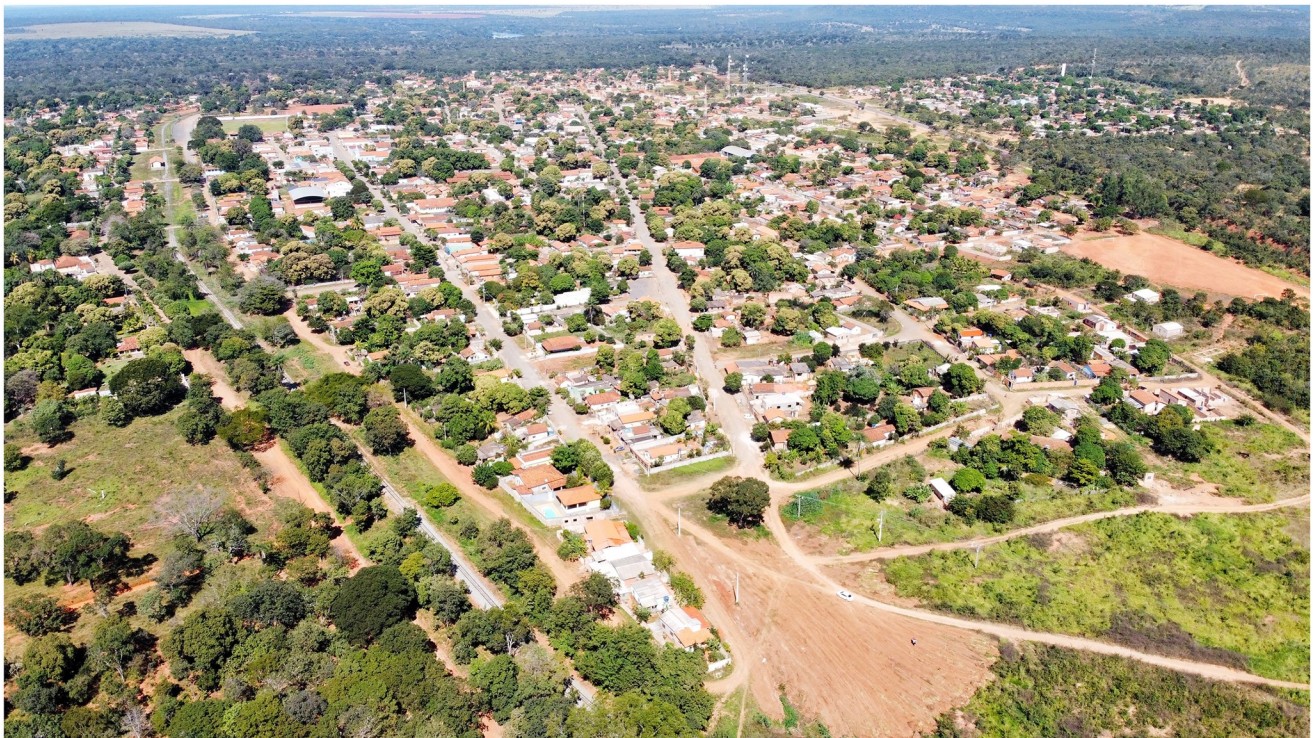

**Fig 2. Aerial view of the urban area of Lassance, Minas Gerais, Brazil.**

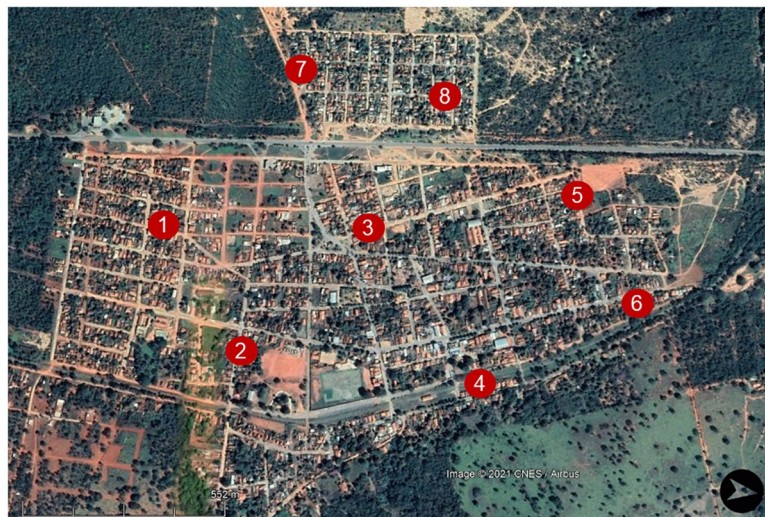

**Fig 3. Distribution of sampling points in the study area of Lassance, Minas Gerais, Brazil.** Red dots with numbers (1–8) identify the sampling points.

## Molecular detection of Leishmania DNA

Total DNA was extracted using a Gentra Puregene KIT (Qiagen, USA), following the manufacturer's protocol, from females individually or in pools for samples larger than 10 individuals of the same species from the same sampling point during the same collection period. A 300–350 bp fragment of the intergenic region of *Leishmania* DNA (Internal Transcribed Spacer q —ITS1) was amplified using the primers LITSR 5´ `CTGGATCATTTTCCGATG` 3´ and L5.8S 5´ `TGATACCACTTATCGCACTT` 3´. The amplified samples were visually analysed by electrophoresis with a 2% agarose gel. To identify *Leishmania* species for positive samples, the amplified product was digested using the enzyme HAEIII (10U/µL), following the manufacturer's recommendations (New England Biolabs, Ipswich, MA, USA). Restriction patterns were analysed on a 4% agarose gel stained with 5µL of ethidium bromide (10U/µL), in comparison with the following reference strains of *Leishmania* spp.: *Le. amazonensis* (IFLA/BR/67/PH8), *Le. braziliensis* (MHOM/BR/75/M2903), *Le. infantum* (MHOM/BR/74/PP75) and *Le. guyanensis* (MHOM/BR/75/M4147).

**Table 1. Characterization of sampling points for sand fly fauna analysis.**

| Sample point | Fruit trees | Shrubs | Lawn | Dog(s) | Chicken(s) | Other animals (n) |
|---|---|---|---|---|---|---|
| 1 | + | + | - | 1 | - | Cat (1) |
| 2 | + | + | - | 1 | - | - |
| 3 | + | + | - | - | - | Horse (1) |
| 4 | + | + | - | 2 | - | Cat (4) |
| 5 | + | + | - | 1 | + | - |
| 6 | + | + | + | - | + | - |
| 7 | + | + | - | - | + | - |
| 8 | + | - | - | 1 | + | Cat (3) |

### Determination of risk areas

The two most abundant species were distributed according to occurrence percentage per sampling point and the result plotted on a map of the municipality to delimit points with the highest concentration of vectors and delimit risk areas.

### Statistical interpretation

Equitability of the sand fly population among the different sampling points, and thus the environments they represent, of the study area was determined by the Pielou Index (J) based on the abundance of each species.

Similarity analysis of the occurrence and relative abundance of sand flies among the different sampling points were determined by Non-Metric Multidimensional Scaling (NMDS) using a dissimilarity matrix calculated according using Jaccard (with species presence/absence data) and Bray-Curtis (with species relative abundance data) indexes.

Statistical analyses were performed using the Vegan package in R software 3.2.4 (R Development Core Team 2016).

### Public equity deposit and ethics statements

Collected sand flies are deposited in the Sand fly Collection of Instituto René Rachou/Fiocruz Minas. This study was performed under a permanent license to collect sand flies issued by IBAMA (number 15237–2).

## Results

A total of 1,611 sand flies of 15 species and eight genera were captured in the urban area of Lassance. The most abundant species were *Lu. longipalpis* and *Ny. intermedia* with 77.09% and 10.06% of all sand fly collected, respectively. The most successful sampling points were point 6 (81.81%) followed by point 4 (11.67%) (Table 2).

The months with the greatest sampling success for collected sand flies were May 2018 (12.97%), November 2018 (16.39%) and January 2019 (31.72%). The months with the lowest average temperatures were May 2018 (av. 23.3˚C), June 2018 (av. 23.3˚C) and July 2018 (av. 22.9˚C) and those with the highest were January 2019 (av. 28.9˚C) and February 2019 (av. 27.8˚C). The lowest averages for relative humidity were August 2018 (av. 57.7%) and September 2018 (av. 54.8%) while the highest were for November 2018 (av. 75.4%), December 2018 (av. 69%) and February 2019 (av. 69.4%) (Fig 4).

A total of 427 female sand flies were subjected to molecular analysis, with 12 samples testing positive for PCR of ITS1, for an infection rate of 2.10% (9/427). All positive samples were submitted to RFLP by HAEIII digestion. We found the *Le. infantum* profile for a *Lu. longipalpis* sample from sampling point 6. Most of the positive samples came from sampling point 6 (n = 4; 44,44%), with the others coming from sampling points 8 (n = 3; 33,33%) and 4 (n = 2; 22,22%) (Table 3).

We used the frequencies of *Lu. longipalpis* and *Ny. intermedia* among sampling points to generate a map of risk areas. Frequency bars represent the frequency of the species at the specific sampling point per all sand flies captured in the study (Fig 5).

The study area had low equitability (J ’ = 0.34), with dissimilar species abundances among the different sampling points and *Lu. longipalpis* being the dominant species. Similarity analysis of sampling points by NMDS based on occurrence and relative abundance of sand fly species per trap explained 74% and 92% of the variation in the data with stress values of 0.08 and 0.06, respectively. These explanation percentages and stress values are considered good, since

**Table 2. Sand flies collected in Lassance, Minas Gerais, Brazil from May 2018 to April 2019.**

| Sand flies | Sample points | | | | | | | | |
|---|---|---|---|---|---|---|---|---|---|
| | **1** | **2** | **3** | **4** | **5** | **6** | **7** | **8** | |
| *Br. avelari* | - | - | - | - | 1 | - | - | - | **1** |
| *Brumptomyia* sp. | - | - | - | 2 | - | 2 | - | - | **4** |
| Cortelezzii complex | 3 | 1 | 2 | 20 | 5 | 10 | 1 | 3 | **45** |
| *Ev. cortelezzii* | - | - | - | 2 | - | - | - | - | **2** |
| *Ev. evandroi* | 1 | - | - | 7 | - | 3 | 6 | - | **17** |
| *Ev. lenti* | 2 | - | 2 | 9 | - | 2 | 5 | 4 | **24** |
| *Ev. sallesi* | 1 | 2 | 3 | 21 | 5 | 15 | 1 | - | **48** |
| *Ev. termitophila* | 1 | 1 | 1 | - | - | - | - | - | **3** |
| *Ev. walkeri* | - | - | - | 1 | - | - | - | - | **1** |
| *Lu. longipalpis* | 4 | 4 | 2 | 77 | 4 | 1143 | 5 | 3 | **1242** |
| *Mt. oliveirai* | - | 2 | - | 2 | 1 | - | - | - | **5** |
| *Mi. quinquefer* | - | - | - | - | - | - | - | 1 | **1** |
| *Ny. intermedia* | 1 | 1 | - | 33 | 1 | 121 | 2 | 3 | **162** |
| *Ny. neivai* | - | - | - | - | - | 13 | - | - | **13** |
| *Ps. brasiliensis* | - | - | - | - | 1 | - | - | - | **1** |
| *Ps. lutziana* | 1 | 1 | 1 | 1 | 2 | - | - | 1 | **7** |
| *Sc. sordellii* | 1 | 2 | 1 | 13 | - | 9 | - | 9 | **35** |
| | **15** | **14** | **12** | **188** | **20** | **1318** | **20** | **24** | **1611** |

they reveal a high probability that the similarity analysis reflects the reality of the studied environments. NMDS revealed a strong tendency for similarity in the composition and relative abundance of sand fly species among the different sampling points in the urban area of Lassance, which suggests that all these environments are favourable for the maintenance of sand fly species communities in the study area (Fig 6).

## Discussion

The diversity of sand fly species found in the present study demonstrates that these insects are adapted to environments with different levels of anthropic interference. The study area has

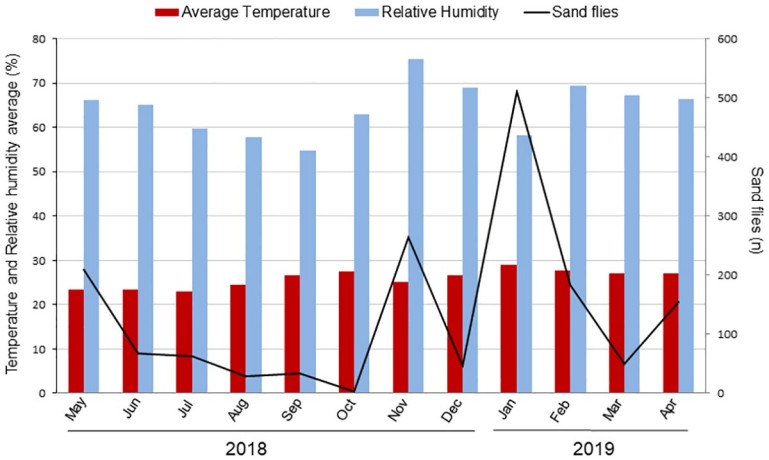

**Fig 4. Average temperature, relative humidity and sand fly abundance during the sampling period in Lassance, Minas Gerais, Brazil.**

**Table 3. PCR IST1 molecular analysis positive samples and *Leishmania* species identification in sand flies collected in Lassance, Minas Gerais, Brazil.**

| Year | Period | Sample point | Sand fly species | *Leishmania* species |
|---|---|---|---|---|
| 2018 | March | 8 | *Lu. longipalpis* | - |
| | June | 6 | *Ny. intermedia* | - |
| | July | 4 | Cortelezzii complex | - |
| | September | 6 | *Lu. longipalpis* | - |
| | | 6 | *Lu. longipalpis* | - |
| 2019 | February | 8 | *Sc. sordellii* | - |
| | March | 4 | Complexo cortelezzii | - |
| | | 6 | *Lu. longipalpis* | *Le. infantum* |
| | April | 8 | *Lu. longipalpis* | - |

characteristics of both peri-urban and rural areas, with more than 70% of the area being the Cerrado biome [15].

Furthermore, all sampling points possess factors (green areas and animal presence) that favour the presence of sand flies. All the sampling points were quite similar with regard to characterization as green areas, with the exception of point 6, which is the only one with grass, and point 8, the one lacking bushes. Even so, it is believed that these variables do not interfere, positively or negatively, with the sampling of sand flies. All sampling points had at least one domestic animal present. The importance of the presence of domestic animals and the level of urbanization in a given area are already known to act on the maintenance and urbanization of diseases such as leishmaniasis in several regions of Brazil [19–23].

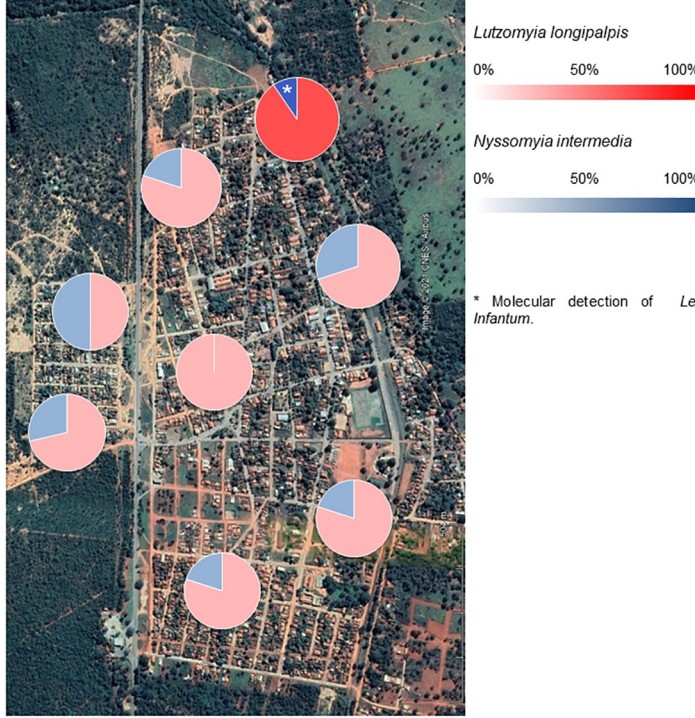

**Fig 5. Risk areas based on distribution of incriminated leishmaniasis' vector sand flies species captured among the sample points in Lassance municipality.**

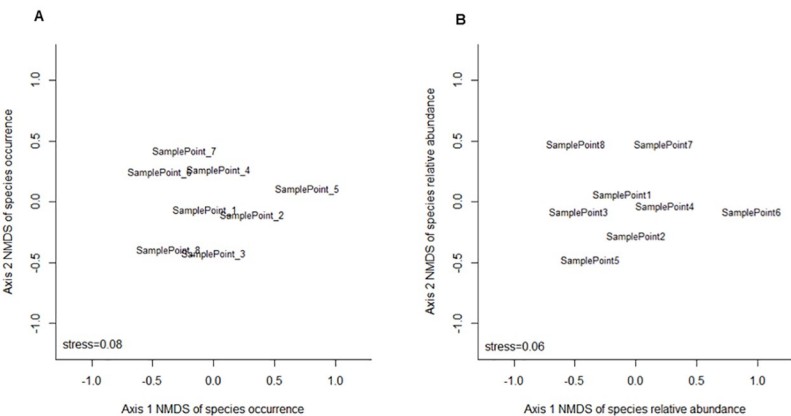

**Fig 6. NMDS analysis of occurrence and abundance of sand flies in relation to sampling points.** A) Plot for NMDS of occurrence data. B) Plot for NMDS of relative abundance data.

The composition of the sand fly fauna in the region of Lassance has been previously analysed with results similar to those of the present study [9,11]. However, no data have been presented regarding the sand fly fauna in the urban area of the city, where a high rate of cases is observed of canine leishmaniasis has been observed (unpublished data). Still, considering that the city of Lassance is an area of transition between urban and rural areas, the diversity of sand flies observed corroborates other studies in the Northern Region of the state of Minas Gerais [20,24,25]. Sampling data revealed the presence of putative vectors, such as *Lu. longipalpis*, *Ev. cortelezzii*, *Ev. sallesi*, *Ny. intermedia* and *Ny. neivai* [26–33], which is of concern because outbreaks of leishmaniasis may arise from this scenario.

The most abundant species in the study, *Lu. longipalpis* (77.09%) and *Ny. intermedia* (10.06%), have great medical importance in the transmission of the etiological agents of visceral and integumentary forms of leishmaniasis and may be involved in the circulation of the parasite in the study area. *Lutzomyia longipalpis* was found distributed among all sampling points and accounted for more than half of all sand flies captured. The sampling point with the greatest capture of *Lu. longipalpis* was point 6, this place corresponded to 81.8% of all sand flies collected and also 92.0% of all *Lu. longipalpis*. This considerable abundance from the same sampling point supports the hypothesis of the existence of local factors that favour such a large number of specimens of *Lu. longipalpis*. However, when comparing the characteristics of sampling point 6, and its location in the study area, to the other sampling points, we could not identify any differential characteristic that could explain such an abundance.

Still, the dominance of *Lu. longipalpis* in the study area probably explains the low value for the equitability index (J' = 0.34), since the great abundance of only one species tends to impact the uniformity of the others. The strong tendency for similarity among the phlebotomine communities of the sampling points may be due to the great dispersion capacity that these insects possess allowing them to reach the different sampling points in the urban area of Lassance, since most species of sand flies were found in all of them at the same relative abundances.

During October 2018, point 6 was under renovation with the dismantling of a chicken coop housing no chickens. Only one specimen *Lu. longipalpis* was collected at point 6 during this period, evidencing the amplified relationship between the presence of chickens and sand fly abundance, as has already been reported in other studies [33–36]. However, three other points (5, 7 and 8) had the presence of chicken coops, yet the number of sand flies collected was much lower than for point 6. Field studies aiming to better interpret the preferred places of

sand flies are needed to achieve a better understanding of the ecology of these organisms. We note that the months with greatest sampling success were followed by periods of higher levels of relative humidity, suggesting a positive relationship between air humidity and sand fly abundance After drier periods, there was a decrease in sampling success. The relationship between relative air humidity and sand fly abundance was also discussed by Teodoro et al [37] and Macedo et al [38]. We observed no relationship between temperature and sand fly abundance, and believe that it did not interfere with sampling success because of the limited variation between the warmest and coldest periods (only av. 5˚C).

Due to limited DNA during the process of individual extraction and of amplified product, only for one sample of those testing positive for PCR of ITS1 was it possible to observe restriction patterns following HAE III digestion. On this occasion, *Le. Infantum* DNA was detected in a sample from *Lu. longipalpis*. In addition, 9 samples tested positive for the detection of *Leishmania* DNA by the ITS1 target. The greater number of positive samples from sampling point 6 probably reflects the greater abundance sampled at the site. Infection of *Lu. longipalpis* by *Leishmania* is already well explained in the literature [26,39,40]. The relationship between the species *Ny. intermedia* and *Leishmania* is also well known in the scientific community, with this species being involved in the spread of cutaneous leishmaniasis [27,28,33,41]. The cortelezzi complex may also be involved in the transmission of the parasites *Le. infantum* and *Le. braziliensis*, however only in one study, also carried out in the region of Lassance, was found promastigotes forms in the mid gut of one species of this complex [30]. The infection rate of 2.10% found in the present study corroborates the molecular detection of *Leishmania* in sand flies mentioned in the work of Tonelli et al. [42]. With respect to *Sc. sordellii*, this species has already been found with DNA from *Leishmania* [43,44], however these reports should be viewed with caution as the genus *Sciopemyia* feeds on cold-blooded animals [45,46].

We separated the occurrences of *Lu. longipalpis* and *Ny. intermedia* among the sampling points and distributes on the map of Lassance, thus creating a distribution of risk areas in the study area. The area of sampling point 6 has the highest risk of transmission of leishmaniasis given the sampled abundance of *Lu. longipalpis* (65.13%) and *Ny. intermedia* (6.73%), and because *Le. infantum* was detected there. In the other regions, the species in question are distributed in approximately 0.6% for *Lu. longipalpis* and 0.09% for *Ny. intermedia*.

The present study represents the first phlebotomine faunal survey carried out in the municipality of Lassance, Minas Gerais. The results obtained demonstrate the circulation of the parasite *Le. infantum*, of vector species and of species with potential for playing a role in the epidemiology of leishmaniasis in Lassance. The strong predominance of *Lu. longipalpis* draws attention to the high risk of transmission and dissemination of new cases of visceral leishmaniasis at the site. Finally, the present study demonstrates the need to conduct more in-depth studies in the field in order to elucidate pertinent questions regarding the preferred places and possible characteristics of the environment that influence sand fly abundance.

## Author Contributions

**Conceptualization:** Gabriel Barbosa Tonelli, Victoria Laporte Carneiro Nogueira, Gabriela Gonçalves Theobaldo, Aldenise Martins Campos, Carina Margonari de Souza, José Dilermando Andrade Filho.

**Data curation:** Camila Binder, Marina Henriques Prado, Gabriela Gonçalves Theobaldo.

**Formal analysis:** Gabriel Barbosa Tonelli, Camila Binder, Marina Henriques Prado, Aldenise Martins Campos, Carina Margonari de Souza, José Dilermando Andrade Filho.

**Funding acquisition:** Carina Margonari de Souza, José Dilermando Andrade Filho.

**Investigation:** Gabriel Barbosa Tonelli, Camila Binder, Victoria Laporte Carneiro Nogueira, Marina Henriques Prado, Gabriela Gonçalves Theobaldo, Aldenise Martins Campos.

**Methodology:** Gabriel Barbosa Tonelli, Victoria Laporte Carneiro Nogueira, Aldenise Martins Campos, Carina Margonari de Souza, José Dilermando Andrade Filho.

**Project administration:** Carina Margonari de Souza, José Dilermando Andrade Filho.

**Supervision:** Carina Margonari de Souza, José Dilermando Andrade Filho.

**Writing – original draft:** Gabriel Barbosa Tonelli.

**Writing – review & editing:** Gabriel Barbosa Tonelli, Camila Binder, Victoria Laporte Carneiro Nogueira, Marina Henriques Prado, Gabriela Gonçalves Theobaldo, Aldenise Martins Campos, Carina Margonari de Souza, José Dilermando Andrade Filho.

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
