## [Decision Letter · Decision Letter 0]

13 Aug 2021

PONE-D-21-17111

The sand fly (Diptera: Psychodidae) fauna of the urban area of Lassance, Northeast Minas Gerais, Brazil.

PLOS ONE

Dear Dr. Andrade-Filho,

Thank you for submitting your manuscript to PLOS ONE. After careful consideration, we feel that it has merit but does not fully meet PLOS ONE’s publication criteria as it currently stands. Therefore, we invite you to submit a revised version of the manuscript that addresses the points raised during the review process.

Please, attempt to incorporate most or all the reviewers' suggestions, made to further improve the manuscript, before we may proceed with a final acceptance of the submission.

We look forward to receiving your revised manuscript.

Kind regards,

Albert Schriefer, M.D., Ph.D.

Academic Editor

PLOS ONE

Journal Requirements:

3.We note that you have referenced (ie. Bewick et al. [5]) which has currently not yet been accepted for publication. Please remove this from your References and amend this to state in the body of your manuscript: (ie “Bewick et al. [Unpublished]”) as detailed online in our guide for authors

Reviewers' comments:

Reviewer's Responses to Questions

**Comments to the Author**

1. Is the manuscript technically sound, and do the data support the conclusions?

Reviewer #1: Yes

Reviewer #2: Yes

2. Has the statistical analysis been performed appropriately and rigorously? 

Reviewer #1: I Don't Know

Reviewer #2: Yes

3. Have the authors made all data underlying the findings in their manuscript fully available?

Reviewer #1: No

Reviewer #2: Yes

4. Is the manuscript presented in an intelligible fashion and written in standard English?

Reviewer #1: Yes

Reviewer #2: No

5. Review Comments to the Author

Reviewer #1: We consider the work suitable for publication. We can raise as a question the lack of statistical support to affirm the relationship between relative humidity and the abundance of collected sandflies. Additionally, we miss a discussion with the literature about the finding of natural infection of Sc. sordelli by Leishmania. Reference indication follows below:

Da-Silva YY, Sales KGS, Miranda DEO, Figueredo LA, Brandão-Filho SP, Dantas-Torres F. Detection of Leishmania DNA in Sand Flies (Diptera: Psychodidae) From a Cutaneous Leishmaniasis Outbreak Area in Northeastern Brazil. Journal of Medical Entomology, Volume 57, Issue 2, March 2020, Pages 529–533.

Reviewer #2: The objective of the study is relevant considering the expansion of visceral leishmaniasis in urban areas.

Abstract:

Line 40: Include the meaning of NMDS.

Introduction: Good

Methods:

Lines 84 and 91 start with the same words, I suggest a slight change.

Line 113: Sacrificed is not a good term.

Others:

Methods suitable for the study.

Good results, contribute to local epidemiological surveillance.

Discussion and conclusion aligned with the objective of the study.

6. PLOS authors have the option to publish the peer review history of their article (what does this mean?). If published, this will include your full peer review and any attached files.

Reviewer #1: No

Reviewer #2: No

---

## [Author Response · Author response to Decision Letter 0]

13 Aug 2021

August 13, 2021

Manuscript PONE-D-21-17111

Dear Albert Schriefer, M.D., Ph.D.,

Thank you for submitting the article and for the reviewers' suggestions.

Below are our responses to Editor's and Reviewers' comments.

Bewick et al. [5] - Our article did not cite this unpublished work.

 Reviewer #1 We consider the work suitable for publication. We can raise as a question the lack of statistical support to affirm the relationship between relative humidity and the abundance of collected sandflies. Additionally, we miss a discussion with the literature on the finding of natural infection of Sc. sordelli by Leishmania. Reference indication follows below:

Da-Silva YY, Sales KGS, Miranda DEO, Figueredo LA, Brandão-Filho SP, Dantas-Torres F. Detection of Leishmania DNA in Sand Flies (Diptera: Psychodidae) From a Cutaneous Leishmaniasis Outbreak Area in Northeastern Brazil. Journal of Medical Entomology, Volume 57, Issue 2, March 2020, Pages 529–533.

 The reference requested in a paragraph on the detection of Leishmania in Sciopemyia sordellii has been included (pages 272-275).

Climatic data were presented in general. The information generated was collected in another municipality (through INMET - Brazilian meteorological agency), which could cause incorrect information. In our next projects, we will carry out these measurements in loco, with devices (thermometers, hygrometers and pluviometer) individualized for each trap.

Reviewer #2: The objective of the study is relevant considering the expansion of visceral leishmaniasis in urban areas.

Abstract:

Line 40: Include the meaning of NMDS. It was included in full.

Methods:

Lines 84 and 91 start with the same words, I suggest a slight change. The sentence has been changed.

Line 113: Sacrificed is not a good term. We removed the term "sacrificed".

Best Regards

Sincerely,

José Dilermando Andrade Filho

Instituto René Rachou

Oswaldo Cruz Foundation

---

## [Decision Letter · Decision Letter 1]

23 Aug 2021

The sand fly (Diptera: Psychodidae) fauna of the urban area of Lassance, Northeast Minas Gerais, Brazil.

PONE-D-21-17111R1

Dear Dr. Andrade-Filho,

We’re pleased to inform you that your manuscript has been judged scientifically suitable for publication and will be formally accepted for publication once it meets all outstanding technical requirements.

Kind regards,

Albert Schriefer, M.D., Ph.D.

Academic Editor

PLOS ONE

Additional Editor Comments (optional):

Reviewers' comments:

Reviewer's Responses to Questions

**Comments to the Author**

1. If the authors have adequately addressed your comments raised in a previous round of review and you feel that this manuscript is now acceptable for publication, you may indicate that here to bypass the “Comments to the Author” section, enter your conflict of interest statement in the “Confidential to Editor” section, and submit your "Accept" recommendation.

Reviewer #2: All comments have been addressed

2. Is the manuscript technically sound, and do the data support the conclusions?

Reviewer #2: Yes

3. Has the statistical analysis been performed appropriately and rigorously? 

Reviewer #2: N/A

4. Have the authors made all data underlying the findings in their manuscript fully available?

Reviewer #2: Yes

5. Is the manuscript presented in an intelligible fashion and written in standard English?

Reviewer #2: Yes

6. Review Comments to the Author

Reviewer #2: Studies with vector insects are increasingly important in different regions of the planet, specially in tropical regions. The article adds important information about sand flies and, consequently, about leishmaniasis. Complex statistical tests are not necessary for the objective of this work.

It's a good article. All comments have been answered.

7. PLOS authors have the option to publish the peer review history of their article (what does this mean?). If published, this will include your full peer review and any attached files.

Reviewer #2: No

---

## [Editor Report · Acceptance letter]

23 Sep 2021

PONE-D-21-17111R1 

The sand fly (Diptera: Psychodidae) fauna of the urban area of Lassance, Northeast Minas Gerais, Brazil. 

Dear Dr. Andrade-Filho:

I'm pleased to inform you that your manuscript has been deemed suitable for publication in PLOS ONE. Congratulations! Your manuscript is now with our production department. 

Kind regards, 

on behalf of

Dr. Albert Schriefer 

Academic Editor

PLOS ONE